# Neurochemical and Behavioral Characterization after Acute and Repeated Exposure to Novel Synthetic Cannabinoid Agonist 5-MDMB-PICA

**DOI:** 10.3390/brainsci10121011

**Published:** 2020-12-18

**Authors:** Aurora Musa, Nicola Simola, Gessica Piras, Francesca Caria, Emmanuel Shan Onaivi, Maria Antonietta De Luca

**Affiliations:** 1Department of Biomedical Sciences, University of Cagliari, 09042 Cagliari, Italy; aurora.musa@unica.it (A.M.); nicola.simola@unica.it (N.S.); g.piras31@studenti.unica.it (G.P.); francesca.caria91@unica.it (F.C.); 2Department of Biology, William Paterson University, Wayne, NJ 07470, USA; OnaiviE@wpunj.edu

**Keywords:** addiction, adolescence, behavior, dopamine, novel psychoactive substances

## Abstract

Since the early 2000s, herbal mixtures containing synthetic cannabinoids (SCs), broadly known as Spice/K2, have been marketed as a legal marijuana surrogate and have become very popular among adolescents. Adolescence is a critical period of development, which is associated with an increased vulnerability to the central effects of drugs. Despite growing concerns about the negative effects of the use of SCs, newly synthetized compounds are increasingly detected in drugs seized by the authorities, posing a serious threat to public health. 5F-MDMB-PICA has been recently detected and classified as a highly potent agonist of CB1 and CB2 cannabinoid receptors. Here, we first investigated the rewarding properties of 5F-MDMB-PICA in C57BL/6 adolescent and adult mice by in vivo brain microdialysis. Data showed that acute administration of a selected dose of 5F-MDMB-PICA (0.01 mg/kg i.p.) stimulates the release of dopamine in the nucleus accumbens shell of adolescent, but not of adult, mice. To further investigate the consequences of repeated exposure to this dose of 5F-MDMB-PICA, a separate group of adolescent mice was treated for 14 consecutive days and evaluated for behavioral abnormalities at adulthood, starting from 7 days after drug discontinuation. Data showed that this group of adult mice displayed an anxiety-like and compulsive-like state as revealed by an altered performance in the marble burying test. Our study suggests an alarming vulnerability of adolescent mice to the effects of 5F-MDMB-PICA. These findings provide a useful basis for understanding and evaluating both early and late detrimental effects that may derive from the use of SCs during adolescence.

## 1. Introduction

Novel Psychoactive Substances (NPS) are a broad variety of synthetic drugs not regulated by current legislation [1]. Preclinical and clinical findings have shown that NPS at doses lower than those of abused psychoactive drugs, such as cannabis, heroin, cocaine, and amphetamines, may produce comparable central effects, but, with more severe consequences ranging from physical to mental health harms, whether through acute intoxication or chronic consumption.

Synthetic cannabinoids (SCs) are currently the most frequently used NPS worldwide. In Europe, by the end of 2019, 179 out of the 731 NPS reported were SCs [2]. Typically sprayed on herbal mixtures or sold in the form of liquids for electronic cigarettes, SCs are commonly known as Spice or K2, usually marketed as marijuana-like drugs and perceived as risk-free by naive users. They act as Synthetic Cannabinoid Receptor Agonists (SCRA) and usually display equal potency at CB1 and CB2 receptors [3,4]. The use of SCs is constantly growing, particularly among adolescents, and is posing major medical and psychiatric risks. Acute and chronic consumption of SCs have been linked to numerous reports of emergency department admissions for adverse health effects, including signs of addiction and withdrawal, and central and peripheral toxicity (e.g., headache, vomiting, paranoia, tachycardia, hypertension, seizures, agitation, psychosis, extreme anxiety/panic, and hallucinations) [5,6,7,8], and even fatalities. As consumers obtain SCs mainly through unknown sources (i.e., the deep and dark web), they are usually not aware of the identity and purity of these substances. This may cause a significant increase in the risk of adverse health effects that may stem from the consumption of derivatives whose biological effects are ill defined, even to medical personnel. Besides their novelty and accessibility, the repeated use of SCs might be attractive to a variety of individuals, such as regular users of cannabis and those who believe that consumption of SCs would not return a positive result in drug testing procedures [2]. The use of SCs is pervasive and their abuse is a growing phenomenon [9,10]. It is, therefore, necessary to thoroughly characterize the pharmacological properties and biological effects of SCs in order to increase the consumers’ awareness of the toxicity that may be associated with the intake of these substances.

5F-MDMB-PICA [Methyl (S)-2-[1-(5-fluoropentyl)-1*H*-indole-3-carboxamido]-3,3-dimethylbutanoate] (Figure 1) is a SC that has been detected in a number of seized branded herbal smoking (or vaping) mixtures, such as the so called ‘Mind Trip’, ‘Devil’, ‘Armageddon’, ‘Trippy Top’, ‘Tropical High’, ‘Astro’, ‘Red Russian’, ‘Supernova’, ‘AK-47’, ‘Dead Man Walking’, and ‘Joker’ [11]. In the USA, 5F-MDMB-PICA has been included in schedule 1 since April 2019, due to the increasing trend reports of intoxications related to its use. For example, medical records from a SCs multiple intoxication case series, occurred within a 10-h period on 15 August 2018 (New Haven, CT, USA) with 109 emergency departments visits as a result of a new batch of K2 entering the local drug supply chain, which showed urine 5F-MDMB-PICA metabolites in 5 out of 21 patients [12]. Moreover, 13 cases related to 5F-MDMB-PICA have been recently reported by the ToxPortal of EWA (Early Warning Advisory, UNODC) [13], which collects fatalities and clinical cases related to the consumption of NPS. These data further highlight the importance of a rapid pharmacological characterization of 5F-MDMB-PICA. Data currently available from some in vitro studies indicate that 5F-MDMB-PICA binds to and activates human CB1 and CB2 receptors at low nanomolar concentrations [14,15,16]. 5F-MDMB-PICA acts as a full agonist at both receptor subtypes with significantly higher potency than delta-9-tetrahydrocannabinol (THC), which is a feature common to other SCs [17,18]. Different functional assays employed revealed that 5F-MBMD-PICA was significantly more potent than THC and other SCRAs [15,16]. In AtT20-FlpIn neuroblastoma cells stably expressing human CB1 receptors, using an assay of receptor-dependent membrane hyperpolarization, 5F-MDMB-PICA, was shown to be a full agonist at the human CB1 receptor with greater potency (EC 50 = 0.45 nM) and efficacy (E max 110%) than THC (EC 50 = 6.77 nM and E max 50%) [19] compared to the high efficacy CB1/CB2 agonist CP55,940 (E max 100% at 1 μM).

Despite the detection of 5F-MDMB-PICA in various Spice/K2-like products, whose main purchasers and users are most likely adolescents and young adults (15–24 years) [20], a satisfactory pharmaco-toxicological characterization of this substance is lacking. In this regard, it is important to highlight that no preclinical studies in vivo have yet been performed to characterize the effects of 5F-MDMB-PICA on the neurocircuits that mediate reward and motivation. Addressing this issue is of great relevance, since it is well established that the neurobiological mechanism of the early stages of addiction are related to an increased activity of the mesolimbic dopamine (DA) system [21] with a particular reference to its projections from the ventral tegmental area (VTA) to the shell of the nucleus accumbens (NAc), which are critically involved in mediating reward and motivational processes [22]. Previous studies by our group showed that the activation of mesolimbic DAergic transmission is an effect that can be observed in rodents after the administration of the prototypical SC JWH-018 [18] and also of second and third generation SCs (i.e., AKB-48, 5F-AKB-48, STS-135, BB-22, 5F-PB22) [17,23,24,25]. Accordingly, we thought that the first fundamental step in the characterization of the central effects of 5F-MDMB-PICA was to determine whether it induced neurochemical effects similar to those elicited by psychoactive substances with rewarding and addictive properties. To this end, we first performed a dose-response curve of the effects of 5F-MDMB-PICA on NAc shell extracellular DA levels by in vivo brain microdialysis in adolescent C57BL/6 mice. Afterward, we wanted to investigate whether an age-related vulnerability to 5F-MDMB-PICA existed. To this end, the only 5F-MDMB-PICA dose that produced a statistically significant effect on the NAc shell DA (0.01 mg/kg i.p.) was chosen to repeat the microdialysis evaluation in a group of adult C57BL/6 mice. Evaluating the age-related effects of 5F-MDMB-PICA is a very relevant issue, since it is well known that SCs are mainly consumed by adolescents and young adults, and that human adolescence is characterized by a prominent level of experimentation including recreational drug use [26,27,28]. Moreover, exposure to drugs during adolescence may be associated with drug dependence or other psychiatric consequences at adulthood [29,30,31,32]. Nevertheless, adolescent drug vulnerability is not widely studied in animal models of repeated drug administration, especially for drugs like 5F-MDMB-PICA that have recently appeared in the market. Therefore, a separate group of mice were administered for 14 consecutive days with 5F-MDMB-PICA at the dose (0.01 mg/kg i.p.) that was found able to increase NAc shell DA levels during mid-adolescence and late-adolescence. At adulthood, Postnatal Day (PND) 58–65, starting from 7 days after drug discontinuation, these mice underwent through a battery of tests suited to disclose the presence of behavioral abnormalities that are indicative of anxiety-like and compulsive-like state, and cognitive deficits that may occur after exposure to SCs during adolescence.

The characterization of the acute and chronic effects of 5F-MDMB-PICA could be useful for better understanding the immediate and late deleterious consequences that may derive from the consumption of this and other SCs during adolescence.

## 2. Materials and Methods

### 2.1. Animals

A total of 48 animals were included in the study. Adolescent (PND 34–53, weight 19–20 g) and adult (PND > 61, weight 30–32 g) male C57BL/6 mice (Charles River, Europe) were employed for in vivo microdialysis experiments and behavioral studies. Mice were housed in groups of six in standard conditions of temperature (21 ± 1 °C) and humidity (60%) under a 12 h/12 h light/dark cycle (lights on at 7:00 a.m.) with food and water available *ad libitum*. All experiments were carried out in accordance with European Council directives (609/86 and 63/2010) and in compliance with the animal policies approved by the Italian Ministry of Health and the Ethical Committee for Animal Experiments (CESA, University of Cagliari). We made all efforts to minimize pain and suffering, and to reduce the number of animals used.

### 2.2. Drugs

5F-MDMB-PICA was purchased from Cayman Chemical Company (Ann Arbor, MI, USA). The drug was dissolved in 0.5% EtOH, 0.5% Tween 80 and 99% saline, and administered intraperitoneally (i.p.) at the doses of 0.001, 0.01, and 0.03 mg/kg in a volume of injection of 10 mL/kg.

### 2.3. Experimental Design Timeline

Neurochemical studies were performed in adolescent and adult mice acutely injected with either 5F-MDMB-PICA (0.001, 0.01, and 0.03 mg/kg i.p. in adolescents, 0.01 mg/kg in adults) or vehicle. Once established that an acute treatment with the intermediate dose significantly affected DA transmission in the NAc shell, a separate group of adolescent mice were injected once a day for 14 consecutive days from PND 38 to PND 51, with either 5F-MDMB-PICA (0.01 mg/kg i.p.) or vehicle. Mice chronically treated were evaluated by a battery of behavioral tests at different time points starting seven days from drug discontinuation (Figure 2). Data evaluation and analysis were performed by experimenters blind to treatment groups.

### 2.4. In Vivo Brain Microdialysis

To determine whether a single administration of 5F-MDMB-PICA (0.001, 0.01, and 0.03 mg/kg i.p. in adolescent, 0.01 mg/kg i.p. in adult) affected DA transmission in the NAc shell, a first set of in vivo brain microdialysis experiments have been performed [33].

#### 2.4.1. Preparation of Microdialysis Probe

Vertical microdialysis probes, with an active dialyzing portion of 1 mm, were prepared with AN69 fibers (Hospal Dasco, Bologna, Italy), as previously described [18].

#### 2.4.2. Surgery

Mice were anaesthetized with isoflurane gas (Merial, Milano, Italy) and maintained under anesthesia using a breathing tube under a scavenging system while placed in a stereotaxic apparatus and implanted with microdialysis probes in the NAc shell (A + 1.4, L ± 0.4 from bregma, V − 4.8 from dura) according to the mouse brain atlas (Paxinos and Franklin, 2008) as previously reported [18,24].

#### 2.4.3. Dopamine Assessment

On the day after surgery, probes were perfused with Ringer’s solution (147 mM NaCl, 4 mM KCl, and 2.2 mM CaCl_2_) at a constant rate of 1 µL/min. Dialysate samples (20 µL) were injected into an HPLC equipped with a reverse phase column (C18 5 μm, Phenomenex, Torrance, CA, USA) and a coulometric detector (Coulochem II, ESA Chelmsford, MA, USA) to quantify DA. The first electrode of the detector was set at +130 mV (oxidation) and the second at −175 mV (reduction). The composition of the mobile phase was: 50 mM NaH_2_PO_4_, 0.1 mM Na_2_-EDTA, 0.5 mM n-octyl sodium sulfate, 15% (*V*/*V*) methanol, and pH 5.5. The sensitivity of the assay for DA was 5 fmoles/sample. After two hours of washing, DA basal levels were evaluated and estimated as the mean of three consecutive samples whose values did not differ more than ±10 percent.

#### 2.4.4. Histology

At the end of the microdialysis experiment, mice were deeply anesthetized and sacrificed. The probes were removed and the brains stored in formalin (8%) for histological examination to verify the correct placement of the microdialysis probe. Brains were cut with a vibratome (Campden Instruments, Leics, UK) in serial coronal slices oriented according to the mouse brain atlas of Paxinos & Watson (1998), and the location of the probes was reconstructed.

### 2.5. Behavioral Tests

In order to determine whether repeated administrations of 5F-MDMB-PICA (0.01 mg/kg i.p., once a day for 14 consecutive days) during adolescence induced persistent behavioral abnormalities at adulthood, such as anxiety-like state, compulsive-like behavioral activity, and memory deficits, we performed a battery of tests consisting of elevated plus maze (EPM), spontaneous alternation behavior (SAB) in a Y-maze, novel object recognition test (NOR), and marble burying (MB), starting at 7 days after 5F-MDMB-PICA discontinuation.

#### 2.5.1. Elevated Plus Maze

The EPM test is widely used to evaluate the presence of anxiety-like states in rodents [34], and has been shown to reliably detect the anxiogenic-like and anxiolytic-like behavioral effects of cannabinoids [35]. In the present study, the EPM was employed to evaluate the possible induction of spatial anxiety-like behavior by 5F-MDMB-PICA. The apparatus was made of white PVC and consisted of two opposite open arms (25 cm L, 5 cm W) and two opposite closed arms (25 cm L, 5 cm W). The latter were enclosed by walls (15 cm H) along their length. The four arms converged on a central square (5 cm × 5 cm), thus, reproducing the shape of a plus sign, and the apparatus was elevated 30 cm from the floor. After 30 min of acclimatization to the experimental room, the test was performed by individually placing each mouse in the central platform facing an open arm, and letting it free to explore the maze for a single 5-min trial. The apparatus was thoroughly cleaned in between each mouse to eliminate olfactory cues. Tests were videotaped and the mice behavior was later scored by an experimenter blind to treatment groups. A mouse was considered inside a specific arm when having all its four paws inside that arm. The percentage of time spent in open arms was used as indicator of an anxiety-like state.

#### 2.5.2. Spontaneous Alternation Behavior in a Y-Maze

The SAB test is widely used to evaluate general cognitive function and spatial memory in experimental rodents [36]. The apparatus was made of black PVC and consisted of three equal arms (40 cm L, 11 cm W, 20.5 cm H), converging onto a central triangular area. Tests were performed by individually placing each mouse in the central area and leaving it free to explore the apparatus for a single 8-min trial. The apparatus was thoroughly cleaned in between each mouse to eliminate olfactory cues. During the test, mice were videotaped and later evaluated by an experimenter blind to treatment groups to determine the number and sequence of arm entries. A mouse was considered inside a specific arm when having all its four paws inside that arm. The percentage of spontaneous alternation was calculated based on the sequence of arm entries, as previously described [37].

#### 2.5.3. Novel Object Recognition Test

The NOR test is widely used to evaluate non-spatial short-term memory in rodents [38]. NOR was performed in polycarbonate cages (25.5 cm L, 19 cm W, and 14 cm H) that were surrounded by a cardboard wall (50 cm H), and had the bottom covered with sawdust. Objects to be discriminated were plastic-made, differed in their shape and colour, and had neither a genuine significance nor emotional valence for mice. The experimental procedure involved three phases: habituation (S0), acquisition (S1), and testing (S2) and mice were always tested individually. Habituation was performed by placing each mouse in the test cage in the absence of any objects and leaving it free to acclimate to the environment for a single 5-min trial. The day after S0, acquisition was performed by placing each mouse in the test cage together with two identical copies of an object (familiar objects) and allowing free object exploration for 3 min. The testing phase took place 60 min after S1. Mice were exposed to one copy of the objects already presented in S1, plus an object that they had never experienced before (novel object). The mice performance was videotaped during S1 and S2 to determine object exploration, which was defined as the animal sniffing, biting, or touching the object. Sitting on and circling around the objects were not considered object exploration. The objects were thoroughly cleaned after each session, to eliminate olfactory cues, and were counter-balanced for status (novel or old) and location (left or right side of the cage). The percentage of time spent in exploring the novel and the old object during S2 were evaluated for each mouse based on the total amount of time spent in exploring both objects during S2.

#### 2.5.4. Marble Burying

The MB is a rodent model of anxiety-like states and compulsive activity with good face validity [39]. The test was conducted in open transparent plastic boxes (54 cm L, 34.5 cm W, 20 cm H) that had the bottom covered with 5 cm of fresh sawdust. Twenty-four standard glass marbles (1.5 cm in diameter, arranged in six rows of four marbles each) were placed uniformly over the bedding. Mice were individually placed in a box and their activity was recorded for 30 min by a video camera placed above the cage. At the end of the session, animals were gently removed from the boxes, and the number of marbles partially (≥67%) and totally (>95%) buried was counted (Anymaze Software, Stoelting, IL, USA). In order to avoid the presence of olfactory cues, bedding was replaced and marbles were thoroughly cleaned in between each animal.

### 2.6. Statistical Analysis

All the numerical data are given as mean ± SEM. Data were analyzed with repeated measures ANOVA (in vivo microdialysis), which was followed by Tukey’s post-hoc test when appropriate, or Student’s *t*-test (basal levels of DA and behavioral tests). Two mice were excluded from the statistical analysis of the NOR test because of a loss of data. Significance was set at *p* < 0.05.

## 3. Results

### 3.1. In Vivo Neurochemical Effects of 5F-MDMB PICA

#### 3.1.1. Microdialysis Studies

Mice basal values of DA in the NAc shell, expressed as fmoles/20 μL sample (mean ± SEM), were: adults 22 ± 4 (*N* = 9), adolescent 21 ± 2 (*N* = 20), and no significant differences between groups were observed (df = 27, t = 0.403; *p* > 0.05) (Student’s *t*-test).

#### 3.1.2. Effects of 5F-MDMB-PICA Administration on DA Transmission in the NAc Shell of Adolescent and Adult Mice

In this set of experiments, we studied the effects of three doses of 5F-MDMB-PICA (0.001, 0.01, and 0.03 mg/kg i.p.) or Vehicle on the extracellular DA levels in NAc shell of adolescent mice. As shown in Figure 3A, the dose-response curve of the effect of 5F-MDMB-PICA on dialysate DA in the NAc shell is bell-shaped. In fact, the intermediate dose of 0.01 mg/kg i.p. significantly increased DA levels with respect to basal values, whereas the higher and lower doses did not. Two-way ANOVA showed a main effect of treatment (F_(3,16)_ = 4.56, *p* < 0.02) and time (F_(6,96)_ = 3.73, *p* < 0.003). Tukey’s post hoc tests applied to the time factor showed a significant increase of dialysate DA in the NAc shell of adolescent mice compared to a respective basal value at 20 and 40 min after treatment.

#### 3.1.3. Role of Age on the NAc Shell DA Stimulation Induced by 5F-MDMB-PICA

In this set of experiments, we studied the differences in the release of DA in the NAc shell elicited by the administration of 5F-MDMB-PICA (0.01 mg/kg i.p.) or vehicle in adolescent and adult mice (Figure 3B). Three-way ANOVA showed a main effect of treatment (F_(1,16)_ = 10.46; *p* < 0.005). In order to evaluate significant differences with respect to basal DA values and between groups, two-way ANOVA of data obtained in adolescent and adult mice treated with 5F-MDMB-PICA (0.01 mg/kg i.p.) was performed. Analysis showed a main effect of age (F_(1,12)_ = 7.52, *p* < 0.02) and time (F_(6,72)_ = 4.73, *p* < 0.0005), and age × time significant interaction (F_(6,72)_ = 3.23, *p* < 0.008). Tukey’s post hoc tests revealed a significant increase of dialysate DA in the NAc shell of adolescent mice compared to respective basal values 20 and 40 min after treatment, and significant differences between adolescents and adults at the same time points.

### 3.2. Behavioral Effects at Adulthood after 5F-MDMB-PICA Repeated Treatment during Adolescence

Adult mice that were subjected to repeated treatment with 5F-MDMB-PICA (0.01 mg/kg, i.p.) during adolescence displayed an altered performance in the MB test. Conversely, the same animals did not show overt behavioral abnormalities when evaluated in the EPM, SAB, and NOR tests.

#### 3.2.1. EPM

Within-group analysis revealed that adult mice treated with either vehicle or 5F-MDMB-PICA as adolescents spent a higher percentage of time in the closed arms than in the open arms of the EPM (Vehicle: df = 9, t = 5.85, *p* < 0.05, 5F-MDMB-PICA: df = 7, t = 15.04, *p* < 0.05) (Figure 4A). However, between-group analysis showed that adult mice treated with either vehicle or 5F-MDMB-PICA as adolescents displayed no significant differences in the percentages of open (df = 16, t = 1.61, *p* > 0.05) and closed (df = 16, t = 0.74, *p* > 0.05) arm exploration. Nevertheless, 5F-MDMB PICA-treated mice displayed non-significant trends toward increased exploration of closed arms and decreased exploration of open arms, compared with vehicle-treated mice. The trend toward increased exploration of closed arms in 5F-MDMB PICA-treated mice persisted when results were expressed as the difference between the percentage of time spent in closed arms and the percentage of time spent in open arms. Moreover, evaluation of the seconds of locomotion performed during EPM exploration revealed no significant differences between mice treated with vehicle (68 ± 2.05) and mice treated with 5F-MDMB PICA (64.88 ± 2.29). This finding indicates that treatment with 5F-MDMB PICA during adolescence did not result in enduring modifications in locomotor activity that may have eventually influenced mice performance in the behavioral tests used in the present study.

#### 3.2.2. SAB

Between-group analysis showed that adult mice treated with either vehicle or 5F-MDMB-PICA as adolescents displayed comparable percentages of SAB in the Y maze (df = 16, t = 0.823, *p* > 0.05) (Figure 4B).

#### 3.2.3. NOR

Within-group analysis revealed that adult mice treated with either vehicle or 5F-MDMB-PICA as adolescents spent a higher percentage of time in exploring the new objects than the old objects (Vehicle: df = 7, t = 3.926, *p* < 0.05, 5F-MDMB-PICA: df = 7, t = 2.327, *p* < 0.05) (Figure 4C). However, between-group analysis showed that adult mice treated with either 5F-MDMB-PICA or vehicle as adolescents displayed no significant differences in the percentages of time spent exploring the new (df = 13, t = 0.11, *p* > 0.05) and old (df = 13, t = 0.15, *p* > 0.05) objects.

#### 3.2.4. MB

*T*-test revealed that adult mice treated with 5F-MDMB-PICA as adolescents buried a significantly higher number of marbles than adult mice treated with vehicle as adolescents (df = 16, t = 3.447, *p* > 0.05) (Figure 4D).

## 4. Discussion

In the present study, we have demonstrated that a selected dose of the latest generation synthetic cannabinoid component of “Spice/K2” drugs 5F-MDMB-PICA significantly stimulated DA transmission in the shell region of the NAc of adolescent, but not adult, mice. Moreover, repeated exposure to 5F-MDMB-PICA during adolescence modified the behavior of adult mice in the MB test, which is used to evaluate the presence of anxious/compulsive-like traits.

The pharmacological property of increasing DA transmission in the NAc shell allows us to relate an unclassified substance, like 5F-MDMB-PICA, was at the beginning of this study to the very well-known drugs with abuse potential, such as heroin, nicotine, cocaine, and THC [22]. Hence, the effects of 5F-MDMB-PICA on NAc shell DA are predictive of reinforcing properties and abuse liability, which is consistent with other findings obtained in our laboratory, showing that 5F-MDMB-PICA is self-administered by adolescent mice (Margiani et al. in preparation) [40]. Notably, the ability of 5F-MDMB-PICA to increase DA transmission in the NAc shell is consistent with previous results, showing a similar effect in rodents treated with JWH-018, which is the prototypical component of synthetic marijuana [18], but also with cannabinoids belonging to the third generation of SCs, such as BB-22, AKB48, 5F-AKB48, STS 135 [18,23,24,25], or to THC itself [41,42,43]. The presence of an inverted U-shaped dose-response effect of 5F-MDMB-PICA on DA release in the NAc shell is in line with the results of previous studies that have characterized the same pattern of response after the administration of other SCs to rodents [18,23,24,25].

A very important finding of the present study is that 5F-MDMB-PICA at the dose of 0.01 mg/kg stimulated NAc shell DA transmission in adolescents but not in adult mice. Notably, this study was performed during mid-adolescence and late-adolescence associated with higher vulnerability to THC rewarding properties in rats [44]. However, it is noteworthy that adult C57BL/6 mice have been previously shown to be very sensitive to the effects that other SCs, such as JWH-018, may elicit on mesolimbic DA transmission [18]. Therefore, we may conclude that the inability of 5F-MDMB-PICA to stimulate NAc shell DA transmission reflects a specific age-dependence of the effects of this drug at the specific dose that is effective in adolescents, that issue deserves further investigation. Additional studies in adolescent and adult mice are currently being designed in our laboratory aimed at further clarifying the dose-response curve of the neurochemical effects of 5F-MDMB-PICA, and at elucidating how these effects may be influenced by gender and strain (Margiani et al., in preparation) [40]. Nevertheless, our data clearly support the hypothesis of an increased vulnerability to the effects of SCs on DA transmission during adolescence, which is in line with earlier findings with other drugs of abuse. Adolescence is a pivotal stage for clinical onset of psychiatric diseases including substance use disorders, especially for alcohol, nicotine, and cannabis [26,27,28]. Consistently, some preclinical studies in rodents have provided evidence of a greater sensitivity of adolescents than adults to the positive rewarding properties of cocaine [45] and amphetamine [46]. In addition, adolescent rats seem to be less sensitive than adults to the aversive properties of stimuli [47] and to the negative effects of withdrawal from nicotine [48,49]. Furthermore, microdialysis studies performed across the different stages (early, mid, and late) of adolescence have revealed that mid-adolescence is the most critical window of vulnerability to the effects that specific drugs of abuse elicit on DA transmission. Notably, mid-adolescent and late-adolescent rats have been shown to be more sensitive to the DA releasing properties of THC in the NAc shell when compared with adults [44], suggesting that the neurochemical effects of psychoactive substances with a cannabinoid-like profile may be particularly evident when they are consumed at mid-adolescence and/or late adolescence. Our results provide support to this view, as they demonstrate the presence of stimulant effects of 5F-MDMB-PICA on mesolimbic DA in mid-adolescent and late-adolescent mice that were no longer evident in adult mice. These age-related differences in the neurochemical effects of 5F-MDMB-PICA could depend on an ongoing endocannabinoid system remodeling during the transition from adolescence to adulthood. In fact, the brain CB1Rs expression reaches the highest values during adolescence while it decreases in adulthood, specifically in cortical regions [50]. Additionally, a fluctuation of anandamide (AEA), 2-arachidonoylglycerol (2-AG), and fatty acid amide hydrolase (FAAH) levels is also observed [51]. These changes can also explain the high sensitivity to natural cannabinoid exposure during adolescence [52,53,54].

In order to better mimic the human pattern of consumption, the dose of 5F-MDMB-PICA that stimulated DA transmission in the NAc shell was used to perform a repeated treatment during the whole mid-adolescence and late-adolescence in a separate group of mice (i.e., from PND 38 to PND 51, 0.01 mg/kg i.p. once a day for 14 consecutive days). Since the long-term consequences of exposure to last generation SCs during adolescence on behavior at adulthood remain poorly understood, adult mice were evaluated in a battery of tests to ascertain the presence of anxiety-like and compulsive-like traits, and of deficits in spatial and non-spatial short-term memory. The results of behavioral experiments indicate that exposure to 5F-MDMB-PICA may elicit persistent detrimental effects on emotional functions. In fact, adult mice repeatedly treated with 5F-MDMB-PICA during adolescence displayed increased burying behavior in the MB test, which is a phenotypical marker thought to reflect the presence of increased anxiety [55,56] and/or of compulsive activity [57]. Nevertheless, the same mice displayed no significant modifications in arm exploration of the EPM, which is a test used to assess the presence of spatial anxiety in rodents. In agreement with a previous study on cannabinoids [35], in mice, aversive behavior was not observed during protracted withdrawal from 5F-MDMB-PICA (i.e., 7 days after drug discontinuation). These discrepant results may be explained by considering the different behavioral significance of the MB and EPM tests, which may lead us to speculate that spatial anxiety is less sensitive to the detrimental effects of 5F-MDMB-PICA, compared with non-spatial anxiety. Alternatively, it may be conceivable that the effects of 5F-MDMB-PICA on spatial anxiety may become overtly manifested when the drug is administered at higher doses or for periods of time longer than those used here, and/or closer to behavioral evaluation compared to what was done here. Similar considerations may apply also to the results showing that exposure to 5F-MDMB-PICA during adolescence did not affect memory function at adulthood, which may appear surprising at first, since previous studies [58] have demonstrated that SCs, such as WIN55,212-2, may modify rodents’ performance in cognitive function, such as the Morris water maze, fear conditioning, reversal learning, and delay discounting tests [58,59]. In this regard, it is important to consider that in the present study we used tests that are devoid of a learning component which, on the contrary, is inherent in the tests (i.e., the Morris water maze and the reversal learning) used in other studies. Neither do the tests used in the present study allow us to evaluate other components of the cognitive domain, such as impulsivity and attention. Accordingly, and also in consideration of the mixed results obtained in studies that evaluated the effects of SCs on cognitive functions, we cannot rule out the possibility that 5F-MDMB-PICA may elicit detrimental effects on the cognitive domain, which could possibly be fully disclosed by using behavioral tests more complex than the ones utilized in the present study.

The results of the present study, in agreement with earlier findings, further underlie the importance of experimental factors such as the age and the timing of drug exposure in the manifestation of the neurochemical and behavioral effects of SCs. In this regard, it is also important to consider that differences in metabolism and/or genetic factors among species are also likely to interfere with the effects of SCs on adolescent neurodevelopment and plasticity, which may be reflected in the presence of neurochemical and behavioral alterations at adulthood. Therefore, the results of the present study are relevant not only to the characterization of the acute and delayed central effects of 5F-MDMB-PICA, but also to further clarify how the vulnerability of the developing central nervous system during adolescence may be a factor that influences the manifestation of the neurochemical and behavioral effects elicited by SCs [60]. The data presented here also suggest the abuse potential of the new synthetic cannabinoid 5F-MDMB-PICA, thus requiring great caution in use. These findings are specifically alarming since adolescents seem to be the most vulnerable subjects to this NPS. Additional pharmacological and toxicological characterizations can improve risk assessment and awareness to care for public health and safety.

## Figures and Tables

**Figure 1 brainsci-10-01011-f001:**
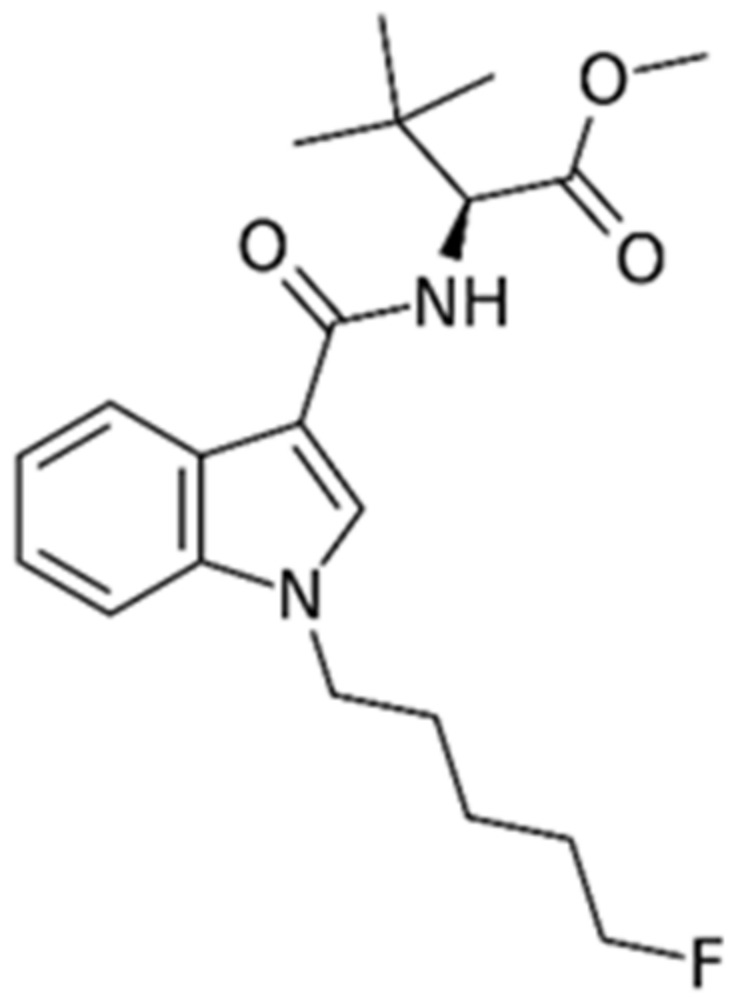
Chemical structure of 5F-MDMB-PICA [Methyl (S)-2-[1-(5-fluoropentyl)-1*H*-indole-3-carboxamido]-3,3-dimethylbutanoate ].

**Figure 2 brainsci-10-01011-f002:**
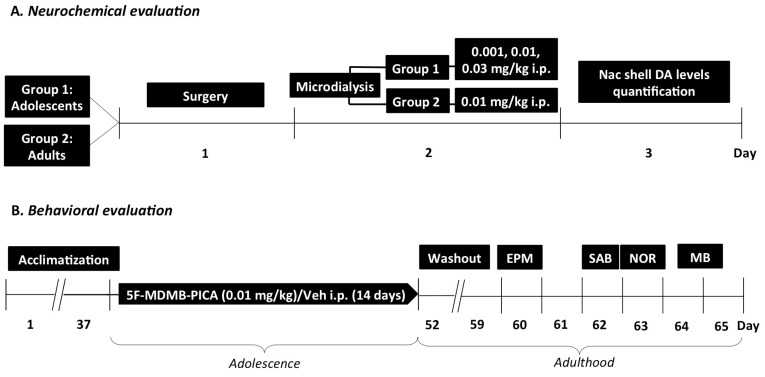
(**A**) Experimental schedule of acute treatment with 5F-MDMB-PICA (0.001, 0.01, and 0.3 mg/kg i.p.) for NAc shell DA levels quantification in adolescent and adult (0.01 mg/kg i.p.) mice by in vivo brain microdialysis. (**B**) Experimental schedule of the repeated treatment with 5F-MDMB-PICA (0.01 mg/kg) or vehicle (intraperitoneal) during adolescence and of behavioral tests at adulthood in male mice. EPM, elevated plus maze. SAB, spontaneous alternation behavior in a Y-maze. NOR, novel object recognition. MB, marble burying tests.

**Figure 3 brainsci-10-01011-f003:**
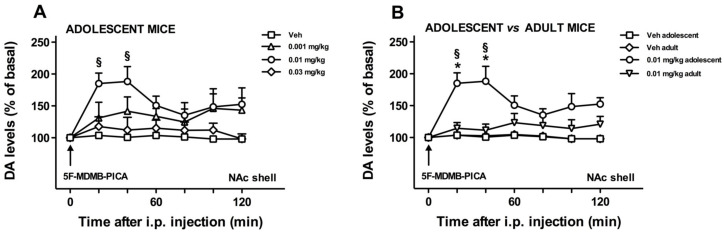
Effects of 5F-MDMB-PICA administration on DA transmission in the NAc shell of adolescent (**A**,**B**) and adult (**B**) mice. Results are demonstrated as mean ± SEM of the changes in DA extracellular levels expressed as the percentage of basal values. (**A**) The arrow indicates the i.p. injection of either Vehicle (squares) or 5F-MDMB-PICA, at the dose of 0.001 mg/kg (ascending triangles), 0.01 mg/kg (circles), 0.03 mg/kg (diamonds). §: *p* < 0.05 vs. basal values (One-way ANOVA; Tukey’s post hoc test). (**B**) The arrow indicates the i.p. injection of either Vehicle or 5F-MDMB-PICA: Vehicle-treated adolescent mice (squares), vehicle-treated adult mice (diamonds), adolescent mice treated with 5F-MDMB-PICA at a dose of 0.01 mg/kg (circles), adult mice treated with 5F-MDMB-PICA at a dose of 0.01 mg/kg (descending triangles), §: *p* < 0.05 vs. basal values. * *p* < 0.05 vs. adult mice (Two-way ANOVA. Tukey’s post hoc test). Vehicle- and 5F-MDMB-PICA-treated adolescent mice are the same as in panel A.

**Figure 4 brainsci-10-01011-f004:**
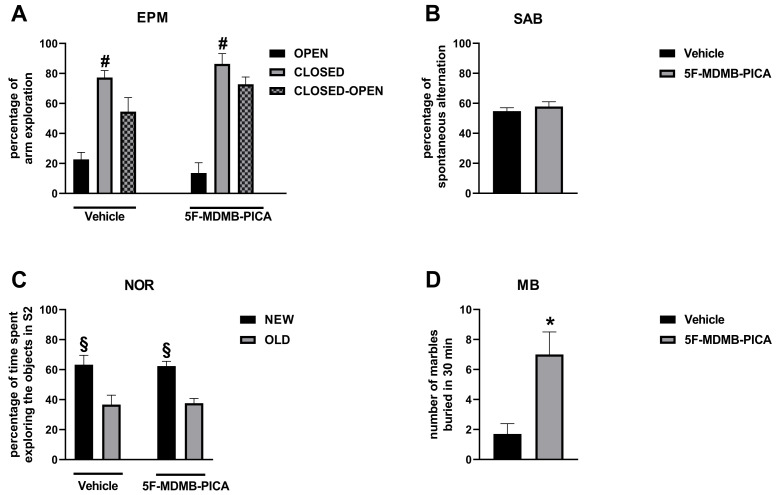
Results of behavioral tests performed in adult mice treated with vehicle or 5F-MDMB PICA (0.01 mg/kg, i.p.) as adolescents. Result are expressed as mean ± SEM of the percentages of (**A**) arm exploration in the EPM. (**B**) SAB in the Y maze. (**C**) Object exploration in the NOR test as well as of the numbers of marbles buried in the EPM, elevated plus maze. (**D**) NOR, novel object recognition. MB, marble burying test. # *p* < 0.05 vs. open arms within each experimental group. § *p* < 0.05 vs. old objects within each experimental group. * *p* < 0.05 vs. mice treated with vehicle as adolescents. Vehicle-treated mice *N* = 8–10; 5F-MDMB PICA-treated mice *N* = 8.

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
