# Peer review of "Neurochemical and Behavioral Characterization after Acute and Repeated Exposure to Novel Synthetic Cannabinoid Agonist 5-MDMB-PICA"

_brainsci, 2020, doi:10.3390/brainsci10121011_

Round 1
Reviewer 1 Report
The present study by Musa et. al. aims to better characterize the acute and chronic effects of 5F-MDMB-PICA administered to adolescent and adult mice. The authors work towards reduced numbers of animals needed for such investigations. I observe the drawback of the manuscript to be the focus on a single NPS SCs and the lack of a direct comparison to other well known SCs. However, the authors do use previously published work to make such comparisons. In general, the manuscript requires moderate revisions for more concise and relevant content. Below some suggestions for improvement are listed:
1. Introduction. The authors assert significantly higher potency of 5F-MDMB-PICA at CB1/CB2 compared to THC but give no potency estimates. Please give numerical values to indicate potency.
2. There are no objective errors in the methods, results and data interpretation however, mice are not "administered"; drugs are administered; so please revise.
3. Methods: Can the actual number of animals used in the study be provided?
4. How was the drug administered in the microdialysis experiment? I.v. catheter or tail vein? Please specify.
5. Results: In Fig 2 it would help if a timeline/time for completion for both neurochemical and behavioural evaluatlion was included in the diagram.
Author Response
Response to Reviewer 1 Comments
General comment: The present study by Musa et. al. aims to better characterize the acute and chronic effects of 5F-MDMB-PICA administered to adolescent and adult mice. The authors work towards reduced numbers of animals needed for such investigations. I observe the drawback of the manuscript to be the focus on a single NPS SCs and the lack of a direct comparison to other well known SCs. However, the authors do use previously published work to make such comparisons. In general, the manuscript requires moderate revisions for more concise and relevant content. Below some suggestions for improvement are listed:
General response: We appreciate the positive feedback regarding the topic and aim of this paper, and we would like to thank Reviewer #1 for his/her critical and substantive feedback. Overall, we feel that the revised manuscript is substantially improved after addressing these comments below. Changes have been highlighted in the text.
Point 1: Introduction. The authors assert significantly higher potency of 5F-MDMB-PICA at CB1/CB2 compared to THC but give no potency estimates. Please give numerical values to indicate potency.
Response 1: In the present version of the manuscript, from lines 72 to 79, more details and numerical value of 5F-MDMB-PICA and THC potency and efficacy at CB1 receptors have been provided as follows: “(…) In AtT20-FlpIn neuroblastoma cells stably expressing human CB1 receptors, using an assay of receptor-dependent membrane hyperpolarization, 5F‐MDMB‐PICA was shown to be a full agonist at the human CB1 receptor with greater potency (EC 50=0.45 nM) and efficacy (E max 110%) than THC (EC 50= 6.77 nM and E max 50%) [19] compared to the high efficacy CB1/CB2 agonist CP55,940 (E max 100% at 1 mM). (…)”
Point 2: There are no objective errors in the methods, results and data interpretation however, mice are not "administered"; drugs are administered; so please revise.
Response 2: revised troughout the manuscript.
Point 3: Methods: Can the actual number of animals used in the study be provided?
Response 3: Done. Total number of animals used are now indicated in section 2.1. Animals (line 118). The numbers of animals used in the neurochemical and behavioral studies are indicated in the sections 3.1.1. (Microdialysis studies) and 3.2. (Behavioural effects at adulthood after 5F-MDMB-PICA repeated treatment during adolescence), respectively.
Point 4: How was the drug administered in the microdialysis experiment? I.v. catheter or tail vein? Please specify.
Response 4: In both neurochemical and behavioural experiments, drugs were administered intraperitoneally (i.p.). This information was already present in the section 2.2. (Drugs) of the previous version of the manuscript, but also written in the abstract, results and discussion. Therefore no additional indication have been made.
Point 5: Results: In Fig 2 it would help if a timeline/time for completion for both neurochemical and behavioural evaluation was included in the diagram.
Response 5: Neurochemical and behavioural studies were performed in 2 different groups of animals. For this reason, in Fig 2, we provided 2 different timelines (i.e. panel A and panel B). Although we stated that neurochemical studies were used also to establish the dose of 5F-MDMB-PICA for behavioural studies, we believe that a single timeline would not be suitable for describing the real progress of the entire experiment.
Reviewer 2 Report
Thank you for sharing the results of your research. Your comprehensive approach demonstrates DA release in the NA, suggesting significant abuse potential, and documents behavioral effects of adolescent administration, in adulthood. It’s a useful model and greatly contributes to the understanding of this novel synthetic cannabinoid.
Introduction: The authors present a rationale for the importance of better understanding of widely-used synthetic cannabinoids. They present a unique rationale for research addressing adolescent administration and future adult models of behavior.
Materials and Methods: Only male mice were used in this research. The battery of behavioral tests was comprehensive. The tests in the battery were explained clearly. Including a measure of general locomotor activity (could be derived from one of the tests as total CM traveled, add rota rod, etc.) would help establish that behavioral changes were not due to locomotor impairment. If you can add even a very basic measure of distance traveled, or % time mobile for one or some of the tests, that would be great to include.
Results
Figure 3: The vehicle data points are too small to see as a hexagon. Using filled circles to represent significant differences is a little confusing in this figure. Is it possible to use a second symbol to denote changes from baseline?
Figure 4: Check the size of font for the panel letter labels. The “c” looks small. In the caption, change the “i, ii…” tables to match the Figure: A, B, C, D. Adding “or vehicle” to the figure title may be helpful. At first glance, it looked like all of the mice in the battery had receive SC at adolescence, and receive either a challenge dose of vehicle or SC at adulthood. Clarifying that small change in the figure title could be helpful to clarify (within the figure) that all mice received vehicle or SC only at adolescence.
3.2.2: Using consistent naming for the Y maze, or the alternating behavior measured, will make this easier for the reader to understand. The Methods present the tests by their names, while the Results present the results of the test using an abbreviation or labeled by the behavior measured by the test. Making the methods and results naming convention more consistent will be helpful.
Line 303: “f” should be “of”
Discussion: I really appreciated the preview of additional research on gender and strain variables - that’s exactly what came to mind for me too! Good anticipation and extension of this line of research. Strain differences may reveal different behavioral profiles (for example, BALB/c mice may exhibit differences in anxiety-like behavior in the EPM or an open field test).
Line 373: “meanly” - is there a lack of understanding in this area? Please clarify word choice.
Not sure if it is possible to conclude that the compound is “unsafe” based on these data, but would certainly warrant additional study and caution in use. Tempering that final claim will be helpful in not going beyond the data presented.
Author Response
Response to Reviewer 2 Comments
General comment: Thank you for sharing the results of your research. Your comprehensive approach demonstrates DA release in the NA, suggesting significant abuse potential, and documents behavioral effects of adolescent administration, in adulthood. It’s a useful model and greatly contributes to the understanding of this novel synthetic cannabinoid.
General response: We appreciate the positive feedback regarding the topic and aim of this paper, and we would like to thank Reviewer #2 for his/her critical and substantive feedback. Overall, we feel that the revised manuscript is substantially improved after addressing these comments below. Changes have been highlighted in the text.
Point 1: Introduction: The authors present a rationale for the importance of better understanding of widely-used synthetic cannabinoids. They present a unique rationale for research addressing adolescent administration and future adult models of behavior.
Materials and Methods: Only male mice were used in this research. The battery of behavioral tests was comprehensive. The tests in the battery were explained clearly. Including a measure of general locomotor activity (could be derived from one of the tests as total CM traveled, add rota rod, etc.) would help establish that behavioral changes were not due to locomotor impairment. If you can add even a very basic measure of distance traveled, or % time mobile for one or some of the tests, that would be great to include.
Response 1: In the present version of the manuscript, from lines 309 to 317, more details on locomotor activity have been provided as follows: “ The trend towards increased exploration of closed arms in 5F-MDMB PICA-treated mice persisted also when results were expressed as the difference between the percentage of time spent in closed arms and the percentage of time spent in open arms. Moreover, evaluation of the seconds of locomotion performed during EPM exploration revealed no significant differences between mice treated with Vehicle (68±2.05) and mice treated with 5F-MDMB PICA (64.88±2.29). This finding indicates that treatment with 5F-MDMB PICA during adolescence did not result in enduring modifications in locomotor activity that may have eventually influenced mice performance in the behavioral tests used in the present study.“
Point 2: Figure 3: The vehicle data points are too small to see as a hexagon. Using filled circles to represent significant differences is a little confusing in this figure. Is it possible to use a second symbol to denote changes from baseline?
Response 2: Figure 3 was modified according to the reviewer’s suggestion. In particular, hexagon has been replaced with squares; data points were increased; significat chenges from baseline have been indicated with a second symbol instead of filled circles.
Point 3: Figure 4: Check the size of font for the panel letter labels. The “c” looks small. In the caption, change the “i, ii…” tables to match the Figure: A, B, C, D. Adding “or vehicle” to the figure title may be helpful. At first glance, it looked like all of the mice in the battery had receive SC at adolescence, and receive either a challenge dose of vehicle or SC at adulthood. Clarifying that small change in the figure title could be helpful to clarify (within the figure) that all mice received vehicle or SC only at adolescence.
Response 3: we have made all the required changes.
Point 4: 3.2.2: Using consistent naming for the Y maze, or the alternating behavior measured, will make this easier for the reader to understand. The Methods present the tests by their names, while the Results present the results of the test using an abbreviation or labeled by the behavior measured by the test. Making the methods and results naming convention more consistent will be helpful.
Line 303: “f” should be “of”
Response 4: Done.
Point 5: Discussion: I really appreciated the preview of additional research on gender and strain variables - that’s exactly what came to mind for me too! Good anticipation and extension of this line of research. Strain differences may reveal different behavioral profiles (for example, BALB/c mice may exhibit differences in anxiety-like behavior in the EPM or an open field test).
Line 373: “meanly” - is there a lack of understanding in this area? Please clarify word choice.
Response 5: we thank the reviewer for the appreciation. “Meanly” has been replaced with “poorly”.
Point 6: Not sure if it is possible to conclude that the compound is “unsafe” based on these data, but would certainly warrant additional study and caution in use. Tempering that final claim will be helpful in not going beyond the data presented.
Response 6: Done, please see lines 437-441.
Reviewer 3 Report
In this manuscript, the authors first determined whether administration of 5-MDMB-PICA (5-MD), a synthetic cannabinoid, produced increases in NAc dopamine in both adolescent and adult mice. They then determined if 14 days of 5-MD injection during adolescence affected various anxiety-like and cognitive behaviors. The authors report that 5-MD produces an inverted U dose response curve for NAc DA release in adolescents, and that same dose (0.01 mg/kg) does not result in DA release in adults. In the behavioral tests, repeated 5-MD exposure only affected marble-burying, although there was a trend towards an increase in anxiety. From this, the authors conclude that adolescent 5-MD exposure may engage reward systems and produce long-term changes in anxiety/compulsive-like behaviors.
This is a timely manuscript because of the prevalence and dangers of less-regulated cannabinoid compounds, especially during adolescence when there may be increased susceptibility to cannabinoid effects. Further, this fits the special issue, the experiments and results are clearly described, and the conclusions are largely justified. Below are several relatively minor issues that should be addressed prior to publication.
1) In the introduction, lines 73-78 regarding bio-transformation/metabolism seem superfluous in the context of this manuscript. While likely important generally (especially potentially active metabolites), I’m not sure of the relevance here. Perhaps moving to the discussion might be more appropriate.
2) The authors chose to only test one dose of 5-MD in adults (0.01, the dose that produced a response in adolescents) and reported no effect. It is then stated in the first paragraph of the results that “5-MDMB-PICA significantly stimulated DA transmission in the shell region of the NAc of adolescent, but not adult, mice” (lines 318-319). However, it is highly possible that adults and adolescents differ in their response to 5-MD dose. It is mentioned in the discussion that this is currently under investigation, but I feel these data should be included here. These results do not feel parametric currently, and even just testing the other two doses used in adolescents would help the narrative that 5-MD produces differential effects on DA in adolescents vs adults. An alternative would be to soften the language used in the discussion (1stparagraph), because currently it is written to suggest that there is no effect in adults, but because only one dose was used, this is not really a justifiable conclusion.
3) The EPM results are intriguing, especially in light of the marble burying. Although the authors report trend effects, I wonder if there would be significant differences if authors also expressed this data as the difference between closed and open % (ie. subtracting the open arm % from the closed arm % in 4A). Also, I did not see animal numbers listed for these experiments, but perhaps these experiments are slightly underpowered.
4) On line 311, the degrees of freedom and t stats seem incorrect, and possibly reversed? A t-stat of 13 would likely be very significant.
5) Line 315, the significance should read “p < 0.05”, but it is written as “p > 0.05”.
Author Response
Response to Reviewer 3 Comments
General comment: In this manuscript, the authors first determined whether administration of 5-MDMB-PICA (5-MD), a synthetic cannabinoid, produced increases in NAc dopamine in both adolescent and adult mice. They then determined if 14 days of 5-MD injection during adolescence affected various anxiety-like and cognitive behaviors. The authors report that 5-MD produces an inverted U dose response curve for NAc DA release in adolescents, and that same dose (0.01 mg/kg) does not result in DA release in adults. In the behavioral tests, repeated 5-MD exposure only affected marble-burying, although there was a trend towards an increase in anxiety. From this, the authors conclude that adolescent 5-MD exposure may engage reward systems and produce long-term changes in anxiety/compulsive-like behaviors.
This is a timely manuscript because of the prevalence and dangers of less-regulated cannabinoid compounds, especially during adolescence when there may be increased susceptibility to cannabinoid effects. Further, this fits the special issue, the experiments and results are clearly described, and the conclusions are largely justified. Below are several relatively minor issues that should be addressed prior to publication.
General response: We appreciate the positive feedback regarding the topic and aim of this paper, and we would like to thank Reviewer #3 for his/her critical and substantive feedback. Overall, we feel that the revised manuscript is substantially improved after addressing these comments below. Changes have been highlighted in the text.
Point 1: In the introduction, lines 73-78 regarding bio-transformation/metabolism seem superfluous in the context of this manuscript. While likely important generally (especially potentially active metabolites), I’m not sure of the relevance here. Perhaps moving to the discussion might be more appropriate.
Response 1: We agree with the reviewer #3. Lines 73-78 of the previous version have been removed.
Point 2: The authors chose to only test one dose of 5-MD in adults (0.01, the dose that produced a response in adolescents) and reported no effect. It is then stated in the first paragraph of the results that “5-MDMB-PICA significantly stimulated DA transmission in the shell region of the NAc of adolescent, but not adult, mice” (lines 318-319). However, it is highly possible that adults and adolescents differ in their response to 5-MD dose. It is mentioned in the discussion that this is currently under investigation, but I feel these data should be included here. These results do not feel parametric currently, and even just testing the other two doses used in adolescents would help the narrative that 5-MD produces differential effects on DA in adolescents vs adults. An alternative would be to soften the language used in the discussion (1stparagraph), because currently it is written to suggest that there is no effect in adults, but because only one dose was used, this is not really a justifiable conclusion.
Response 2: Following the comments of this reviewer, the language used in the discussion has been softened. In addition, the fact the only one dose was used in adults was indicated throughout the different sections of the manuscript. We agree with the fact that adults most likely respond to higher doses of 5-MD, when compared to adolescents. However, our data on the dose–response curve of adults are too preliminary to be included in the present paper.
Point 3: The EPM results are intriguing, especially in light of the marble burying. Although the authors report trend effects, I wonder if there would be significant differences if authors also expressed this data as the difference between closed and open % (ie. subtracting the open arm % from the closed arm % in 4A). Also, I did not see animal numbers listed for these experiments, but perhaps these experiments are slightly underpowered.
Response 3: In the present version of the manuscript, from lines 309 to 317, more details on differences in behavioural data of control or treated animals have been provided as follows: “ The trend towards increased exploration of closed arms in 5F-MDMB PICA-treated mice persisted also when results were expressed as the difference between the percentage of time spent in closed arms and the percentage of time spent in open arms. Moreover, evaluation of the seconds of locomotion performed during EPM exploration revealed no significant differences between mice treated with Vehicle (68±2.05) and mice treated with 5F-MDMB PICA (64.88±2.29). This finding indicates that treatment with 5F-MDMB PICA during adolescence did not result in enduring modifications in locomotor activity that may have eventually influenced mice performance in the behavioral tests used in the present study.“
Number of animals used in the behavioural studies are now indicated in the legend of Fig. 4.
Total number of animals used are now indicated in section 2.1. Animals (line 118).
Point 4: On line 311, the degrees of freedom and t stats seem incorrect, and possibly reversed? A t-stat of 13 would likely be very significant.
Response 4: Done. The degrees of freedom and t stats have been corrected.
Point 5: Line 315, the significance should read “p < 0.05”, but it is written as “p > 0.05”.
Response 4: Done.